# Analysis of Frailty in Geriatric Patients as a Prognostic Factor in Endovascular Treated Patients with Large Vessel Occlusion Strokes

**DOI:** 10.3390/jcm10102171

**Published:** 2021-05-18

**Authors:** Marlena Schnieder, Mathias Bähr, Mareike Kirsch, Ilko Maier, Daniel Behme, Christian Heiner Riedel, Marios-Nikos Psychogios, Alex Brehm, Jan Liman, Christine A. F. von Arnim

**Affiliations:** 1Department of Neurology, University Medical Center Göttingen, 37073 Göttingen, Germany; mbaehr@gwdg.de (M.B.); ilko.maier@med.uni-goettingen.de (I.M.); jliman@gwdg.de (J.L.); 2Department of Geriatrics, University Medical Center Göttingen, 37073 Göttingen, Germany; mareike.kirsch@med.uni-goettingen.de (M.K.); christine.arnim@med.uni-goettingen.de (C.A.F.v.A.); 3Department of Neuroradiology, University Hospital of Magdeburg, 39120 Magdeburg, Germany; daniel.behme@med.uni-goettingen.de; 4Department of Neuroradiology, University Medical Center Göttingen, 37073 Göttingen, Germany; christian.riedel@med.uni-goettingen.de; 5Department of Interventional and Diagnostical Neuroradiology, Clinic for Radiology and Nuclearmedicine, University Hospital Basel, 4031 Basel, Switzerland; marios.psychogios@usb.ch (M.-N.P.); alex.brehm@usb.ch (A.B.)

**Keywords:** stroke, frailty, elderly patients, hospital frailty risk score, mechanical thrombectomy

## Abstract

Frailty is associated with an increased risk of adverse health-care outcomes in elderly patients. The Hospital Frailty Risk Score (HFRS) has been developed and proven to be capable of identifying patients which are at high risk of adverse outcomes. We aimed to investigate whether frail patients also face adverse outcomes after experiencing an endovascular treated large vessel occlusion stroke (LVOS). In this retrospective observational cohort study, we analyzed patients ≥ 65 years that were admitted during 2015–2019 with LVOS and endovascular treatment. Primary outcomes were mortality and the modified Rankin Scale (mRS) after three months. Regression models were used to determine the impact of frailty. A total of 318 patients were included in the cohort. The median HFRS was 1.6 (IQR 4.8). A total of 238 (75.1%) patients fulfilled the criteria for a low-frailty risk with a HFRS < 5.72 (22.7%) for moderate-frailty risk with an HFRS from 5–15 and 7 (2.2%) patients for a high-frailty risk. Multivariate regression analyses revealed that the HFRS was associated with an increased mortality after 90 days (CI (95%) 1.001 to 1.236; OR 1.112) and a worse mRS (CI (95%) 1.004 to 1.270; OR 1.129). We identified frailty as an impact factor on functional outcome and mortality in patients undergoing thrombectomy in LVOS.

## 1. Introduction

Treatment of older people can be a challenge for health care systems. An aging population leads to a higher frequency of age related diseases such as dementia, cancer or stroke, often in patients with multimorbidity [1]. But hospital admission and even therapies can be a cause of harm for some older people [2]. Analyses of frailty can help to identify those patients. Frailty is described as a decline of function in multiple organ systems linked to aging and an increased risk of poor outcome [3]. Recently, a novel frailty score based on Tenth Revision of the international classification of disease (ICD-10) diagnostic codes was developed and proven to be capable of identifying patients which are at high risk of adverse outcomes [4]. In total, the score consists of 109 ICD-codes. The authors created a points system, where a certain number of points are awarded for each ICD-10 code and added together to create the final frailty risk score. ICD-10 codes with the highest impact are Dementia in Alzheimer’s disease, Hemiplegia, Alzheimer’s disease followed by sequelae of cerebrovascular disease and other signs involving the nervous and musculoskeletal systems, including a tendency to fall. The score has several advantages since it is easy to calculate based on the medical history of the patients with a low interrater variability [4]. Frailty, analyzed via the Hospital Frailty Risk Score (HFRS) has been shown to be correlated with poor outcomes, for example after transcatheter valve therapies [5], catheter ablation of atrial fibrillation [6], heart failure [7,8] as well as acute myocardial infarction [7]. In stroke patients, pre-stroke frailty seems to be associated with a shorter survival [9] and patients with stroke are more likely to be classified as frail [10]. Furthermore, pre-existing comorbidities in stroke are associated with a higher short-term and long-term mortality [11] and it is associated with an attenuated improvement following stroke thrombolysis [12]. But to date, there are no data regarding the impact of frailty on the efficacy of mechanical thrombectomy. Since mechanical thrombectomy has become the standard of care for large vessel occlusion stroke (LVOS) patients after publication of the first five randomized trials in 2015 [13], understanding the mechanisms influencing the outcome has been a challenge. Time from onset of stroke to treatment as well as high Alberta Stroke Program Early CT scores (ASPECTS) is crucial for a favorable outcome [14,15,16].

It is known that increasing age is associated with poor outcomes [13]. Octagenarians and Nonagenarians treated by mechanical thrombectomy have a higher mortality and morbidity than younger patients. Still, successful recanalization leads to a better neurological outcome and a lower mortality in these patients [17,18,19].

Thus, it would be helpful to implement indicators or scores which are of prognostic value in patients undergoing thrombectomy. The aim of this study was to examine the outcome of elderly patients suffering from LVO with regard to frailty.

## 2. Materials and Methods

### 2.1. Study Design, Setting and Study Population

We conducted a retrospective observational cohort study at the University Medical Center in Göttingen by using linked clinical and health administrative databases from 2015 to 2019. This included the stroke database, which we analyzed for elderly patients ≥ 65 years being admitted with LVOS and endovascular treatment. LVOS was defined as a stroke due to an occlusion of the carotid artery, middle cerebral artery in the M1 segment or proximal M2 segment, anterior cerebral artery, posterior cerebral artery or basilar artery. The trial was registered and approved prior to inclusion by the ethics committee of the University of Medicine Göttingen (Ethikkommission der Universitätsmedizin Göttingen (No: 13/7/15An)). Written consent was obtained by all participants or their legally authorized representatives.

### 2.2. Study Outcomes

Primary outcomes of patients were measured by the three months mortality rate as well as the modified Rankin Scale (mRS) after three months. A good outcome was defined as a mRS from 0–2 and a poor outcome as mRS from 3–6.

### 2.3. Data Sources

The prospectively derived stroke database contains data of patients with LVOS undergoing mechanical thrombectomy in the University Medical Center in Göttingen during 2015–2019. The collected data of the stroke database included neurological features such as the National Institute of Health Stroke Scale (NHISS), mRS at discharge and after three months, as well as neuroradiological characteristics such as the Alberta stroke program early CT score (ASPECTS) and the modified thrombolysis in cerebral infarction scale (mTICI). NHISS and mRS were assessed by an experienced neurologist, ASPECTS and mTICI by a senior neuroradiologist. Δ-NIHSS was calculated for each patient as the difference between NIHSS at admission and NIHSS at discharge. For the three-month follow up, patients were examined in person. A telephone interview was made in case the patient was not able to come to the hospital. Furthermore, baseline characteristics such as age and gender of the patients were collected, as well as the average length of stay, rate of pneumonia, rehospitalization rate and mortality after three months. To analyze frailty, the Hospital Frailty Risk Score (HFRS) was calculated for each patient of the database based on the International classification of disease (ICD)-10 codes at time of admission of the patients using the pre-morbid condition of the patients including all data available at the timepoint of stroke admission, including previous admissions. The acute stroke symptoms were not included into the score. The HFRS is a recently developed and validated score to measure frailty [4]. Moreover, individuals were categorized as low (<5), intermediate (5–15) or high risk (>15) for frailty based on previously published cut-off points [4]. Patients in the intermediate-risk and high-risk categories were defined as frail. Apart from HFRS, Elixhauser and Charlson comorbidity indices were calculated for each patient based upon diagnoses of the patients at discharge. Both indices have been reported to be a predictor for mortality [20,21].

### 2.4. Statistical Analysis

For descriptive statistics, continuous variables are presented in means with a standard deviation or a median with an interquartile range. Categorial variables are demonstrated as counts and percentages. Outcomes and the influence of different HFRS risk-categories on hospital stay and pneumonia were assessed using a chi-squared or Kruskal-Wallis-Test as appropriate. To analyze outcomes and mortality of patients after three months, a logistic regression analysis was performed. Univariable logistic regression models were used to identify factors associated with a statistical probability (*p* < 0.001) on the outcome and mortality of the patients and then were included into the multivariable logistic regression model. HFRS, TICI-Scale, the Elixhauser- as well as the Charlston-Comorbitity Index, age, hours ventilation, rate of pneumonia, NHISS at admission and discharge, delta-NIHSS, iv-rtPA, gender, hemicraniectomy, mRS at discharge, ASPECTS, intracranial hemorrhage and time from onset to recanalization were run as univariate models. Results were considered statistically significant when *p* < 0.05. Since there was evidence of a threshold phenomenon, the association of HFRS and mortality was assessed using a segmented linear regression model, as implemented in the R package “segmented” [22]. All statistical analysis was performed using IBM SPSS Statistics vs. 26 (IBM, Armonk, NY, USA), except from regression analysis and c-statistics, which were performed in R.

## 3. Results

Of the 655 patients of our stroke database, 410 patients had complete follow-up data and 318 fulfilled the inclusion criteria (LVOS with mechanical thrombectomy) with an age ≥65 years. The median age of the patients was 80.1 years (IQR 9.58), the majority of patients were female (60.4%) and 40 (12.6%) suffered from pneumonia. The clinical characteristics are presented in Table 1.

The median HFRS was 1.6 (IQR 4.8). When we calculated the HFRS risk categories, 238 (75.1%) patients met the criteria for low-risk with a HFRS < 5, while 73 (22.7%) fulfilled the criteria for moderate-risk with a HFRS from 5–15 and 7 (2.2%) patients for high-risk. Regarding the neurological characteristics, the median NIHSS on admission was 15 (IQR 10) and 8 (IQR 19) at discharge. The median mRS after 90 days was 4 (IQR 5) and 120 patients (37.7%) had died after 90 days. A total of 109 (34.3%) patients had a favorable outcome with a mRS from 0–2, whereas 209 (65.7%) of the patients had an unfavorable outcome with an mRS 4–6. The neuroradiological characteristics of the patients are listed in Table 2.

Frail patients, defined by a HFRS ≥ 5, were older than non-frail patients (83.8 (IQR 9.6) vs. 78.9 (IQR 9.6); *p* < 0.001) but there was no significant difference in age between patients with a moderate or high frailty risk (83.8 (IQR 9.9) vs. 84.2 (IQR 7.2); *p* = 0.753). Detailed information about the differences in the frailty groups are listed in Table 3 and Table 4.

There was no significant difference in mortality at discharge, but there was a significant association between mortality and frailty after 90 days in the logistic regression analysis. In the univariate models, we found that the likelihood of mortality after 90 days significantly increased with HFRS (*p* < 0.001; CI (95%) 1.053 to 1.1183; OR 1.16) as well as the likelihood of an unfavorable neurological outcome (*p* < 0.001; CI (95%) 1.069 to 1.248; OR 1.155). The C-statistics of the model on mortality after 90 days were 0.6293. Multivariate analyses revealed that along with age, the mTICI scale, Δ-NIHSS, ASPECTS and HFRS (*p* = 0.020; CI (95%) 1.018 to 1.240; OR 1.24) showed a significant relationship with the likelihood of mortality after 90 days, as shown in Table 5.

Plotting the relationship between HFRS and the rate of mortality after 90 days suggested a threshold phenomenon, as seen in Figure 1.

There was a steep increase in mortality up to a frailty score of three, afterwards the gradient flattened out until it reached a plateau at about a frailty score of 15. No significant difference was detectable in the gradient of the curve.

When dividing into the three frailty-risk categories, in-hospital death in the low-risk group was 18.5% (44), in the moderate-risk group 25.0% (18) and in the high-risk group 14.3% (1) (*p* = 0.448). After 90 days, the mortality rate was 33.2% (79) in the low-risk group, 50.0% (36) in the moderate-risk group and 71.4% in the high-risk group (5); (*p* = 0.005).

Furthermore, there was a significant influence of the HFRS (*p* = 0.029; CI (95%) 1.012 to 1.254; OR 1.127) along with age, mTICI, ASPECTS, age, the Elixhauser Comorbidity Index and **Δ**-NIHSS in the multivariate analysis on neurological outcomes of the patients, as shown in Table 6.

After dividing into the three frailty categories, patients in a low frailty risk category were more likely to have a favorable outcome than those in a moderate or high frailty risk category; this reached statistical significance (95 (39.9%) vs. 14 (19.4%) vs. 0 (0%); *p* < 0.001) and can be seen in Figure 2.

Frail patients, with a HFRS ≥ 5, did not secondary complications suffer significantly more often such as pneumonia (27 (11.3%) vs. 11 (15.3%) vs. 1 (14.3%); *p* = 0.664) and their rate of mechanical ventilation was similar to patients with a low frailty score (130 (54.6%) vs. 36 (50.0%) vs. 4 (57.1%); *p* = 0.776). The length of stay in the hospital did not differ between patients with a low, moderate or high frailty risk (10 (IQR 11) vs. 10.5 (IQR 8) vs. 9 (IQR 5) days; *p* = 0.656). No patient in the high-risk frailty group exceeded the maximum length of stay (0.29 (±2.413) vs. 0.1 (±0.118) vs. 0 (±0); *p* = 0.776). With respect to all three frailty groups, the total number of patients exceeding the g-DRG calculated maximum of stay was rather small, as can be seen in Figure 3.

A similar length of stay is also reflected in the renumeration for the patients. Renumeration was similar without a significant difference in the three different risk groups (low-risk 18525.83 EUR (IQR 7900.15 EUR) vs. moderate-risk 22418.75 EUR (IQR 7285.41 EUR) vs. high-risk 19903.06 EUR (IQR 10070.45 EUR); *p* = 0.401).

## 4. Discussion

In this monocentric cohort, more than 75% of all endovascular treated strokes were older than 65 years and our study could demonstrate that frailty, as assessed by the HFRS, has an impact on mortality after mechanical thrombectomy in large vessel occlusion stroke. Frail patients have a significantly higher mortality rate after three months than non-frail patients. Compared to previous studies on frail patients with stroke [18], our study provides a very distinct and highly relevant subgroup of strokes, namely LVO induced strokes treated with thrombectomy. We applied the HFRS as a well evaluated frailty score, and were able to correct for multiple confounders due to our comprehensive data set.

In an aging population, frailty and the associated treatment risk is of emerging interest. Multiple scores been developed to predict mortality in patients, such as the Charlson Comorbidity Index and Elixhauser Comorbidity Index [20,21]. However, in our study, the Elixhauser Comorbidity Index and Charlson Comorbidity Index were not able to predict mortality in thrombectomized patients.

The ICD-10 based hospital frailty risk score is based on administrative data and therefore is an easy and accessible score to measure frailty, possibly enabling the physician to calculate the risk for adverse events prior to hospital admission and enabling consultation of patients or relatives.

Previously this score has been used to evaluate outcome after transcatheter valve therapies [5]. In contrast to TAVI procedures, the magnitude of the effect in thrombectomized patients is lower than those undergoing TAVI (HR TAVI 3.1, OR thrombectomy: 1.2).

Moreover, frailty is not only a predictor of mortality in patients, additional to age [13], ASPECTS [15] and other parameters such as time [14]; frailty is another parameter to predict a neurological outcome of mechanical thrombectomy in large vessel occlusion stroke. Frail patients have a worse neurological outcome after 90 days compared to non-frail patients. Only 20% of the medium risk group achieve a favorable outcome after 3 months (mRs 0–2; Figure 1). The effect on neurological outcome might be influenced by premorbid disabilities, since frail patients suffer more often from dementia and premorbid stroke [23] than non-frail patients. Although more detailed analyses are warranted to better understand the possible influence of distinct premorbid diseases on disability and, hence on outcome, further data on premorbid disability beyond specific diseases might also help to address this in future studies.

Apart from mortality and neurological outcomes, we did not find a significant difference in secondary complications or prolonged stays in hospital between different frailty groups. This is in contrast to previous data of a general hospital population [24] and a population after degenerative spine operations [25], in which an association between frailty and prolonged hospitalization had been shown. But medical interventions in stroke patients, leading to a prolonged hospital stay such as hemicraniectomy are rarely performed in frail patients. Moreover, it may be possible that other secondary complications like acute renal failure and septicemia are not treated extensively in high-risk frailty patients, as it is more likely that frail patients were sent back to care facilities in a palliative regime after disabling stroke. Another possibility is that the cut-offs differentiating between low, moderate and high-risk groups may not be accurate for stroke patients. This is indicated by the different curves of the association between mortality and HFRS in our study comparing to the original work [4]. Since HFRS was not intended to describe adverse outcomes in a specific stroke population, future studies are needed to validate the cut-offs in stroke patients.

One strength of our study is the complete data set without missing data, which minimizes a bias.

Another advantage of the HFRS is that it can be implemented automatically in hospital information systems. This refers to patients for whom the necessary administrative data are already available at the time of a new stroke admission. Because the ICD-10 code is routinely recorded electronically, determination of the HFRS can be automatically embedded in the hospital’s electronic medical record and even has the potential to be programmed into frailty-attuned clinical decision support systems. Having the HFRS be automatically available at hospital admission may avoid the challenges of implementing manual scores such as the Clinical Frailty Scale and improve standardization of frailty assessment. This may have additional potential, particularly in stroke patients, where every second counts and acute disability (e.g., hemiplegia, aphasia) may complicate clinical judgement of premorbid frailty status on admission.

Whether the HFRS, a purely clinical frailty assessment or a combination of both is better suited to predict outcome in frail stroke patients needs to be clarified in further studies. In addition, further studies are needed to determine whether a frailty assessment can guide and optimize clinical care at the individual patient level.

Our study has several limitations. One limitation, not only of our study but also the HFRS, is the lack of complete administrative patient data at admission, which may lead to a misclassification bias. Several diagnoses, for example unspecified fall or care diagnoses involving the use of rehabilitation procedures which are part of the HFRS, are not always documented in patients’ medical history, leading to an incomplete and therefore lower score. This is not only a shortcoming of our study but a general problem for the HFRS, in particular when used in a clinical setting. In addition, not all ICD-scores used to calculate the HFRS are used in the German reimbursement system.

Another limitation, leading to a possible selection bias, is that we only included patients who underwent mechanical thrombectomy and not those with LVOS without treatment. This was because frail patients are more likely to be excluded from mechanical thrombectomy compared to non-frail patients due their premorbid disabilities. Therefore, the rather positive clinical outcome results of the frail group may be overestimated. Especially in the high-risk frailty group, the number of patients is rather low. Therefore, it may be possible that our study underpowered this patient subgroup. This problem could be addressed by applying larger patient cohorts, e.g., from multicentric registries, to further explore the effects of strokes on high-risk frailty.

## 5. Conclusions

Frailty leads to a higher mortality and morbidity in patients with large vessel occlusion, even in medium frail patients, only 20% reach a favorable outcome. Therefore, identifying frail patients and stratifying risk categories using the HFRS may help to communicate with patients and families about the incidence of potential outcomes.

## Figures and Tables

**Figure 1 jcm-10-02171-f001:**
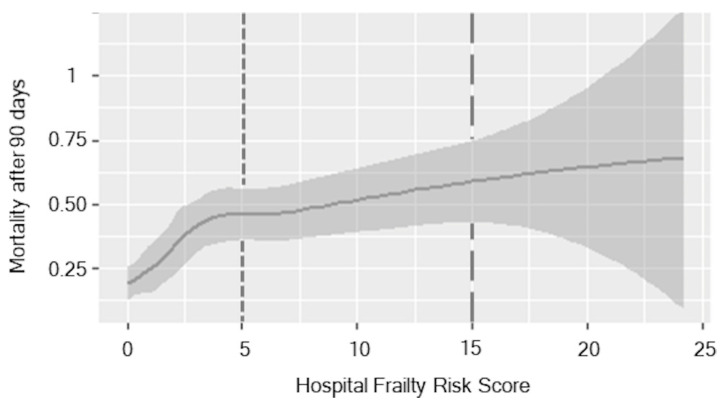
Association of HFRS over mortality after 90 days. The different HFRS risk categories are divided by the grey lines. The first line is the boundary between the low- and moderate-frailty risk category, the second line between moderate- and high-frailty risk. The grey shade is indicating the standard deviation.

**Figure 2 jcm-10-02171-f002:**
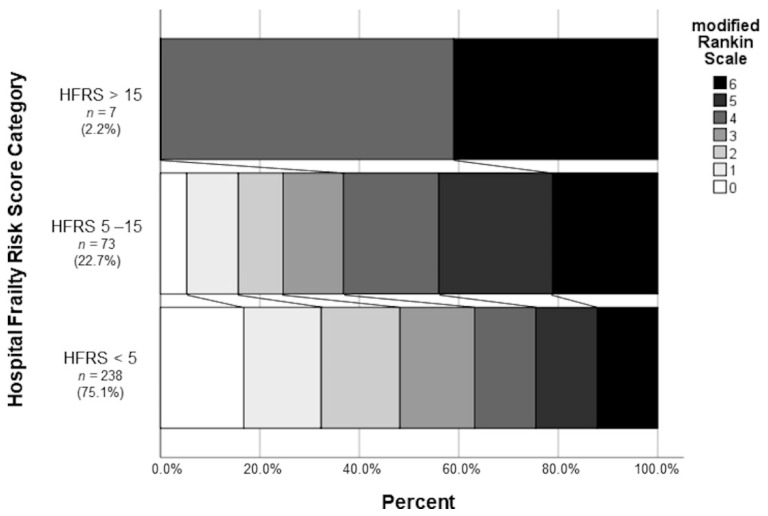
Proportion of the modified Rankin Scale after mechanical thrombectomy as a percentage of the different HFRS categories.

**Figure 3 jcm-10-02171-f003:**
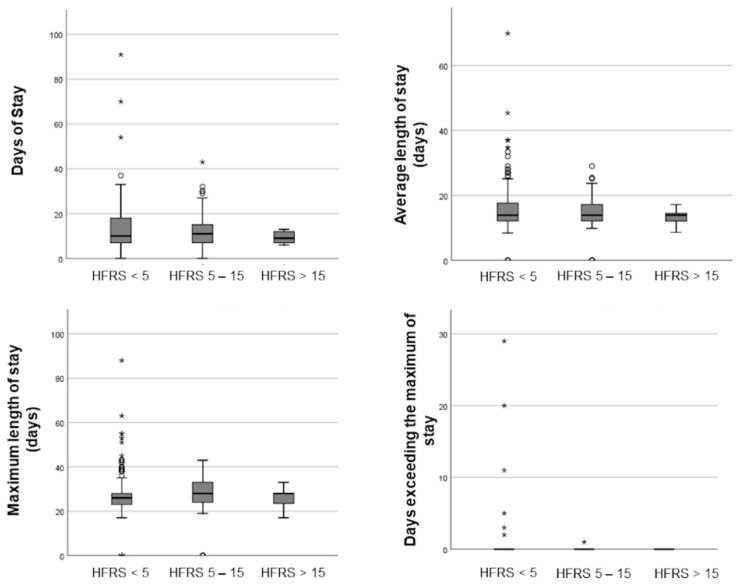
Boxplots of the influence of the HFRS on the length of stay in hospital. A white dot is indicating an outlier (3rd quartile + 1.5 interquartile range or 1st quartile − 1.5 interquartile range) and a * an extreme outlier (3rd quartile + 3 interquartile range or 1st quartile − 3 interquartile range).

**Table 1 jcm-10-02171-t001:** Clinical characteristics.

Clinical Characteristics	Total *n* = 318
Age [median (IQR)]	80.1 (IQR 9.58)
Female [*n* (%)]	192 (60.4%)
Pneumonia [*n* (%)]	40 (12.6%)
NIHSS at admission [median (IQR)]	15.0 (IQR 10)
NIHSS at discharge [median (IQR)]	8 (IQR 19)
mRS at discharge [median (IQR)]	4 (IQR 3.75)
mRS at 90 days [median (IQR)]	4 (IQR 5)
good outcome (mRS 0–2) [*n* (%)]	109 (34.3%)
HFRS [median (IQR)]	1.6 (IQR 4.8)
low frailty risk (<5) [*n* (%)]	238 (75.1%)
moderate frailty risk (5–15) [*n* (%)]	73 (22.7%)
high frailty risk (>15) [*n* (%)]	7 (2.2%)
Charlson comorbidity index [median (IQR)]	4 (IQR 6)
Elixhauser comorbidity index [median (IQR)]	9 (IQR 13)
Hemicraniectomy [*n* (%)]	11 (3.5%)
intravenous thrombolysis [*n* (%)]	187 (58.8%)
in hospital death [*n* (%)]	63 (19.8%)
mortality rate after 90 days [*n* (%)]	120 (37.7%)

NIHSS: National Institute of Health Stroke Score, mRS: modified Rankin Scale, HFRS: Hospital Frailty Risk Score.

**Table 2 jcm-10-02171-t002:** Neuroradiological characteristics.

Neuroradiological Characteristics	*n* = 318
door-to-groin time [min (IQR)]	50 (31)
Time from onset to treatment [min (IQR)]	110 (70)
Time from onset to recanalization [min (IQR)]	231 (210)
periprocedural subarachoid hemorrhage [*n* (%)]	32 (10.1%)
intracerebral hemorrhage [*n* (%)]	39 (12.4%)
mTICI scale	
0	24 (7.6%)
1	6 (1.9%)
2a	29 (9.2%)
2b	75 (23.7%)
2c	50 (15.8%)
3	132 (41.8%)
Occlusion side	
Proximal internal carotid artery	11 (3.5%)
Carotid-T	56 (17.6%)
M1-branch of MCA	152 (47.8%)
M2-branch of MCA	54 (17%)
Basilar artery	32 (10.1%)
ACA	4 (1.3%)
PCA	7 (2.2%)
ASPECTS [median (IQR)]	8 (2)

mTICI scale: modified thrombolysis in cerebral infarction scale, MCA: middle cerebral artery, ACA: anterior cerebral artery, PCA: posterior cerebral artery, ASPECTS: Alberta stroke program early CT score, IQR: interquartile range. a,b,c: a part of the scale.

**Table 3 jcm-10-02171-t003:** Clinical characteristics of the different frailty groups.

Clinical Characteristics	HFRS < 5	HFRS 5–15	HFRS > 15	*p*-Value
age [median (IQR)]	78.9 (9.6)	83.8 (9.6)	84.2 (7.2)	*p* < 0.001
Female [*n* (%)]	137 (57.6%)	50 (68.5%)	5 (71.4%)	*p* = 0.206
Pneumonia [*n* (%)]	27 (11.3%)	11 (15.3%)	1 (14.3%)	*p* = 0.664
NIHSS at admission [median (IQR)]	15 (9)	14 (8)	17 (6)	*p* = 0.254
NIHSS at discharge [median (IQR)]	7 (16)	12 (35)	15 (37)	*p* = 0.052
mRS at discharge [median (IQR)]	3 (4)	4 (4)	5 (3)	*p* = 0.027
mRS at 90 days [median (IQR)]	4 (2)	5 (2)	6 (5)	*p* < 0.001
good outcome (mRS 0–2) [*n* (%)]	95 (39.9%)	14 (19.4%)	0 (0%)	*p* < 0.001
Charlson comorbidity index [median (IQR)]	4 (2)	5 (3.5)	5 (2)	*p* < 0.001
Elixhauser comorbidity index [median (IQR)]	9 (10.5)	15 (14)	13 (10)	*p* = 0.005
Hemicraniectomy [*n* (%)]	9 (3.8%)	2 (2.7%)	0 (0%)	*p* = 0.811
intravenous thrombolysis [*n* (%)]	137 (57.6%)	45 (51.6%)	5 (71.4%)	*p* = 0.652
in hospital death [*n* (%)]	44 (18.5%)	18 (25.0%)	1 (14.3%)	*p* = 0.448
mortality rate after 90 days [*n* (%)]	79 (33.2%)	36 (50.0%)	5 (71.4%)	*p* = 0.005

IQR: interquartile range, NIHSS: National Institute of Health stroke score, mRS: modified Rankin Scale.

**Table 4 jcm-10-02171-t004:** Neuroradiological characteristics of the different frailty groups.

Neuroradiological Characteristics	HFRS < 5	HFRS 5–15	HFRS > 15	*p*-Value
Door-to-groin time [min (IQR)]	50 (31)	47 (40)	54 (50)	*p* = 0.572
Onset to recanalization time [min (IQR)]	228 (198)	241.5 (293)	219.5 (103)	*p* = 0.697
onset to treatment time [min (IQR)]	107.5 (66)	115 (86)	140	*p* = 0.798
periprocedural subarachoid hemorrhage [*n* (%)]	23 (10.1%)	8 (11.1%)	1 (16.7%)	*p* = 0.855
intracerebral hemorrhage [*n* (%)]	32 (13.6%)	7 (9.7%)	0 (0%)	*p* = 0.410
TICI [*n* (%)]				*p* = 0.676
0	18 (7.6%)	5 (6.8%)	1 (14.3%)	
1	5 (2.1%)	1 (1.4%)	0 (0%)	
2a	17 (7.2%)	11 (15.1%)	1 (14.3%)	
2b	57 (24.2%)	15 (20.5%)	3 (42.9%)	
2c	38 (16.1%)	11 (15.1%)	1 (14.3%)	
3	101 (42.8%)	30 (41.1%)	1 (14.3%)	
Occlusion site				*p* = 0.039
Proximal ACI	10 (4.2%)	1 (1.4%)	0 (0%)	
Carotid-T	46 (19.3%)	8 (11%)	2 (28.6%)	
M1-branch of MCA	114 (47.9%)	38 (52.1%)	0 (0%)	
M2-branch of MCA	31 (13%)	18 (24.7%)	5 (71.4%)	
Basilar artery	26 (10.9%)	6 (8.2%)	0 (0%)	
ACA	3 (1.3%)	1 (1.4%)	0 (0%)	
PCA	6 (2.5%)	1 (1.4%)	0 (0%)	
ASPECTS	8 (2)	9 (1)	9 (3)	*p* = 0.165

IQR: interquartile range, mTICI: modified thrombolysis in cerebral infarction scale, ACI: internal carotid artery, ACA: anterior cerebral artery, PCA: posterior cerebral artery, ASPECTS: Alberta stroke program early CT score. a,b,c: a part of the scale.

**Table 5 jcm-10-02171-t005:** Multivariate logistic regression analysis of Influence on mortality after 90 days; HFRS: Hospital Frailty Risk Score, mTICI scale: modified thrombolysis in cerebral infarction scale, ASPECTS: Alberta Stroke Program Early CT Score, NIHSS: National Institute of Health Stroke Score, Δ-NIHSS: difference between the NIHSS at admission and discharge.

Mortality after 90 Days	Odds Ratio	95% Confidence	Interval	*p*-Value
HFRS	1.124	1.018	1.240	0.020
Age (years)	1.159	1.090	1.232	<0.001
mTICI scale	0.760	0.581	0.993	0.044
ASPECTS	0.740	0.576	0.951	0.019
Δ-NIHSS	0.868	0.835	0.903	<0.001

**Table 6 jcm-10-02171-t006:** Multivariate logistic regression analysis of neurological outcome after 90 days measured by mRS.

Poor Neurological Outcome (mRS 3–6)	Odds Ratio	95% Confidence	Interval	*p*-Value
HFRS	1.127	1.012	1.254	0.029
Age	1.077	1.023	1.135	0.005
ASPECTS	0.584	0.450	0.758	<0.001
mTICI scale	0.696	0.526	0.921	0.011
Elixhauser Comorbidity Index	1.074	1.028	1.122	0.001
Δ-NIHSS	0.897	0.857	0.939	<0.001

HFRS: Hospital Frailty Risk Score, ASPECTS: Alberta Stroke Program Early CT Score, mTICI scale: modified thrombolysis in cerebral infarction scale, mRS: modified Rankin Scale, NIHSS: National Institute of Health Stroke Score.

## Data Availability

The data presented in this study are available on request from the corresponding author due to the need to respect the privacy of the patients.

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
