# Peer review of "Analysis of Frailty in Geriatric Patients as a Prognostic Factor in Endovascular Treated Patients with Large Vessel Occlusion Strokes"

_jcm, 2021, doi:10.3390/jcm10102171_

Round 1
Reviewer 1 Report
Thank you for the opportunity to review this manuscript.
Summary:
Schnieder et al. performed a retrospective cohort study on the value of the hospital frailty risk score (HFRS) for outcome prediction in 318 patients aging 65 years or older who suffered from endovascular treated (thrombectomy) large vessel occlusion stroke. In a multivariate regression analysis, they found a rather low increased Odds ratio for 90-days-mortality (OR 1.1) and worse modified rankin scale after three month.
The study is pleasantly written, well presented, and adequately edited. In my view, however, there are gross conceptual flaws:
Looking at the distribution of frailty determined by HFRS in the entire cohort, it is striking that it is very low overall (median 1.6). Only 2.2% are classified as "very frail." There are two major problems with this study. First, it is doubtful that an ICD-10 based assessment of frailty adequately reflects the clinical condition. Second, and supporting the first criticism, the result of the multivariate logistic regression with an OR of 1.1 at a 95% of 1.0 to 1.2 (p 0.05) is of highly questionable relevance. Thus, the study does not answer the question whether frailty predicts the outcome of patients after thrombectomy at all. If one follows the results of the multivariate analysis, one would indeed rather conclude that frailty plays (almost) no role in prognosis. The Conclusion also claims that the HFRS is easy to collect. This statement is not realistic for everyday care reality - at least in Germany.
Author Response
Thank you for carefully revising our manuscript. However, we disagree with your opinion that frailty, measured by the HFRS is not relevant. First of all, the percentages of the distribution for different frailty groups in our paper are similar as those in the literature (Eckart et. al., BMJ open). We agree, that the HFRS can only be as good as the quality of the administrative data. However, it has been shown, that the HFRS reflects the clinical condition of patients in different health care systems in Europe (Eckart et al., BMJ open), the United States (Kundi et al.; JAMA Cardiol.) or Canada (McAlister et al., BMJ Qual. Saf). Moreover, the lack of complete administrative data leads rather to underestimation of the impact of frailty on stroke treatment, we would like to emphasis with our project the importance of easily accessible patient data, as we believe, that research should not accept existing boundaries as limitation. Hopefully, in future it will be easier to calculate the HFRS in Germany as well, for example with upcoming tools such as the health care card. However, we attenuated the argument that the HFRS is easy to collect in an emergency setting.
Reviewer 2 Report
I thank the authors for letting me review this interesting manuscript.
In this manuscript, the authors address the prognostic factor of frailty in geriatric patients with large vessel occlusion treated with endovascular therapy. They introduce frailty measured by the ICD 10 based hospital frailty risk score (HFRS) as a predictor for mortality and morbidity in patients ≥65. This manuscript is of high relevance, since with rising life expectancy and increasing care dependency a substantial number of elderly suffer from ischemic stroke and factors for risk calculations are much needed. The manuscript is well written and of good structure.
I would like to point out several issues to the authors:
General:
- When expanding the use of endovascular treatment to frail or elderly people, who had not been part of the RCTs, it seems unlikely to be less effective regarding protecting cerebral tissue. The question is rather the safety of the intervention and therefore the overall benefit. The latter is well addressed in this manuscript, but it would be of interest to the reader to know more about safety in these frailty groups, e.g. by showing rate of intracerebral haemorrhage and periprocedural complications.
Methods:
- (69) The authors have excluded all patients that did not receive endovascular treatment, despite LVOS. This is likely to lead to a selection bias. Frail patients are more likely to be excluded from EVT treatment on clinical grounds, most likely because of premorbid disabilities some of which may be underrepresented in the HFRS, compared to non-frail patients. Therefore, the rather positive clinical outcome results of the frail group may be overestimated. The authors should point this out in the limitation section.
Results:
- To better understand the direct comparison between the three frailty groups, it would be of interest to see baseline characteristics for each group. Also, if available, process times and premorbid functional disability (e.g. measured with the modified Rankin Scale), being important influences on clinical outcome, would be of interest to the reader.
- (117) 318 patients were mentioned, who fulfilled inclusion criteria, while Table 1 shows a total number of 352. Could the authors please clarify this difference (also regarding some of the percentages in table 1)
- (128) Likewise, mortality rate and total number seems to differ between table 1 and text
- Table 2: modified Rankin Scale at admission is difficult to interpret as this scale was not developed for in-hospital use. Questions like: is the patient able to perform as before? or even: is the patient able to walk with assistance? cannot be reliably answered in the hyperacute setting. I suggest to remove this item from analyses and rather use admission NIHSS instead.
Discussion
- I would appreciate the authors discussing why they think that high HFRS is associated with unfavourable outcome in this particular group of patients. May this be just because of a higher rate of premorbid disability in this group (see Eckart et al, BMJ open 2019; 9(1))?
Author Response
In this manuscript, the authors address the prognostic factor of frailty in geriatric patients with large vessel occlusion treated with endovascular therapy. They introduce frailty measured by the ICD 10 based hospital frailty risk score (HFRS) as a predictor for mortality and morbidity in patients ≥65. This manuscript is of high relevance, since with rising life expectancy and increasing care dependency a substantial number of elderly suffer from ischemic stroke and factors for risk calculations are much needed. The manuscript is well written and of good structure.
I would like to point out several issues to the authors:
General:
- When expanding the use of endovascular treatment to frail or elderly people, who had not been part of the RCTs, it seems unlikely to be less effective regarding protecting cerebral tissue. The question is rather the safety of the intervention and therefore the overall benefit. The latter is well addressed in this manuscript, but it would be of interest to the reader to know more about safety in these frailty groups, e.g. by showing rate of intracerebral hemorrhage and periprocedural complications.
Answer: We thank the reviewer for bringing this important issue to our attention. We added the rate periprocedural subarachnoid hemorrhage and intracerebral hemorrhage to the neuroradiological characteristics (table 2). Furthermore, we analyzed the frailty groups regarding the rate of complications (table 4). There was a trend towards more periprocedural subarachnoid hemorrhages and less intracerebral hemorrhages with increasing frailty scores, which was not significant. This may be due to the low sample size of the complications.
Methods:
- (69) The authors have excluded all patients that did not receive endovascular treatment, despite LVOS. This is likely to lead to a selection bias. Frail patients are more likely to be excluded from EVT treatment on clinical grounds, most likely because of premorbid disabilities some of which may be underrepresented in the HFRS, compared to non-frail patients. Therefore, the rather positive clinical outcome results of the frail group may be overestimated. The authors should point this out in the limitation section.
Answer: This is a very important point and we added this issue to the limitations section in the discussion.
“Another limitation, leading to a possible selection bias, is that we only included patients who underwent mechanical thrombectomy and not those with LVOS without treatment. Because frail patients are more likely to be excluded from mechanical thrombectomy compared to non-frail patients due their premorbid disabilities. Therefore, the rather positive clinical outcome results of the frail group may be overestimated.” (266-270)
Results:
- To better understand the direct comparison between the three frailty groups, it would be of interest to see baseline characteristics for each group. Also, if available, process times and premorbid functional disability (e.g. measured with the modified Rankin Scale), being important influences on clinical outcome, would be of interest to the reader.
Answer: Following the reviewer’s suggestion, we added the baseline characteristics and neuroradiological characteristics for each group (table 3 and 4), including the times. We added mRS at discharge and mRS at 90 days in table 3. Unfortunately, regarding the premorbid functional disability, measured by the modified Rankin scale, we only have very incomplete data.
- (117) 318 patients were mentioned, who fulfilled inclusion criteria, while Table 1 shows a total number of 352. Could the authors please clarify this difference (also regarding some of the percentages in table 1)
- (128) Likewise, mortality rate and total number seems to differ between table 1 and text
Answer: We carefully revised table 1 as well as the other numbers for the correct values and percentages.
- Table 2: modified Rankin Scale at admission is difficult to interpret as this scale was not developed for in-hospital use. Questions like: is the patient able to perform as before? or even: is the patient able to walk with assistance? cannot be reliably answered in the hyperacute setting. I suggest to remove this item from analyses and rather use admission NIHSS instead.
Answer: Following the reviewers’ suggestion, we removed the modified Rankin Scale at admission from our analysis.
Discussion
- I would appreciate the authors discussing why they think that high HFRS is associated with unfavorable outcome in this particular group of patients. May this be just because of a higher rate of premorbid disability in this group (see Eckart et al, BMJ open 2019; 9(1))?
Answer: Thank you very much for this important remark. According to Fried et al. (2004) there is a complex relationship between frailty, comorbidity and disability. In our study, we focused on frailty, but also included two relevant comorbidity indices. However, we can only estimate the degree of disability from specific diagnoses. Unfortunately, regarding the premorbid functional disability, measured by the modified Rankin scale, we only have very incomplete data. We added this point to the discussion.
Frail patients have a worse neurological outcome after 90 days than non-frail patients. Only 20% of the medium risk group achieve a favourable outcome after 3 months (mRs 0-2; Fig.1). The effect on neurological outcome might be influenced by premorbid disabilities, since frail patients suffer more often from dementia and premorbid stroke[20] than non-frail patients. Although more detailed analyses are warranted to better understand the possible influence of distinct premorbid diseases on disability and, hence on outcome, further data on premorbid disability beyond specific diseases might help to address this in future studies. (235-242)
Reviewer 3 Report
A nice paper on the impact of frailty on thrombectomy outcomes. The subject is of clinical interest and although the results are modest and rather predictable.
- Please define an LVO in this study context and present the distribution of the treated vessels.
- ”Hospital Frailty Risk Score (HFRS) was calculated for each patient of the database based on the International classification of disease (ICD)-10 codes at time of admission” Recenty it has been suggested that the data from two previous admissions should also be used: https://www.thelancet.com/journals/lanhl/article/PIIS2666-7568(21)00004-0/fulltext Could this be implemented here?
- ”Univariable logistic regression models were used to identify factors associated with a high impact (p<0.001)” Two grievances here: 1) the p-value does not signify ”high impact”, only statistical probability; 2) which parameters were run as univariate models? (and how were multiple comparisons with?)
- Were treatment times (OTT, door-to-groin) available for analysis? It would be surprising if they were not as the data were derived from a prospective stroke registry and known to be important. These should be included.
- mRS at admission is not really something one would expect to be reported. The reader will easily confuse this with pre-stroke mRS which it apparently is not here (meadian mRS 5). mRS is not an acute parameter and the authors should stick to NIHSS for this.
- On the other hand, pre-stroke mRS scores would be very important for putting the functional outcome data into context. After all, frailty is strongly associated with functional status and the poor outcome might be partially explained by the already poor functional status before the stroke.
- That could also explain the peculiar dissonance between the results on functional outcome and mortality with frailty being the most important predictor of the first but the least important predictor of the latter in the analyses. Mortality is a far more robust and unequivocal endpoint and also probably closer to the truth in this case considering the subject.
- Since the study period includes both pre- and post-DAWN periods is it possible to analyse patients treated within six hours and patients treated in longer time frames also separately?
- ”However, the results did not reach statistical significance.” Please do not use deterministic wording. It is not a priori clear that the result would be heading that way but just fell short for some reason or other – a point rather nicely emphasised by the behavior of the curve.
- ”The length of stay in the hospital did not differ between 178 patients with a low, moderate or high frailty risk (10 (IQR 11) vs. 10.5 (IQR 8) vs. 9 (IQR 179 5) days; p= 0.656). Moreover, the average length of stay was similar without any significant difference (13.9 (IQR 6) vs. 13.9 (IQR 5) vs. 13.9 (IQR 4); p = 0.763)” This does not make any sense. Please clarify.
- Indeed, frailty has not been assessed (or at least published) as a predictor of thrombectomy outcome – but advanced age has and, as also the current results show, these tend to go hand in hand. The authors should also discuss previous thormbectomy studies on octo- and nonagenarians. (Btw, there seems to be an Editorial coming up in Age&Ageing soon on the subject)
- ”But still almost 20% of the medium risk group achieve a favorable outcome after 3 months” So, four out of five patients in this group did not. And the results in the high risk group were uniformly poor, almost dismal. These points should be emphasised as they are important for clinical decision-making.
- ”Frailty measured by the ICD 10 based hospital frailty risk score is an easy and accessible method, bases on administrative data” I would not call a 109-point list ”easy and accessible” to a clinician. For sure, an EHR appendage could probably calculate it but this leads to the question of the quality of administrative data (as also noted in the discussion) and the method of calculating the score. No, I think these results rather suggest that frailty is also important in thrombectomy patients and should be evaluated clinically.
- Furthermore, its effect seems surprisingly modest here but this might result from the problems with the administrative data already noted? One more good reason to favor clinical assessment and also make a new study along these lines.
- The style of writing is cumbersome in both the Introduction and the Discussion especially as both are written as singular long paragraphs (the same applies to a lesser extent to Results). These should be revised for the benefit of the reader. Moreover, there are incomplete sentences and repetition. The text should go through a careful round of text editing with fresh eyes.
Author Response
A nice paper on the impact of frailty on thrombectomy outcomes. The subject is of clinical interest and although the results are modest and rather predictable.
- Please define an LVO in this study context and present the distribution of the treated vessels.
Answer: We defined LVOS in the methods section and added the distribution of the treated vessels into table 1.
“LVOS was defined as a stroke due to an occlusion of the carotid artery, middle cerebral artery in the M1 segment or proximal M2 segment, anterior cerebral artery, posterior cerebral artery or basilar artery.” (70-72)
- ” Hospital Frailty Risk Score (HFRS) was calculated for each patient of the database based on the International classification of disease (ICD)-10 codes at time of admission” Recenty it has been suggested that the data from two previous admissions should also be used: https://www.thelancet.com/journals/lanhl/article/PIIS2666-7568(21)00004-0/fulltext Could this be implemented here?
Answer: Thank you for this comment, we used all data available at the timepoint of stroke admission. This means, that all data, including previous admissions, were used. We explained this in more detail in the methods section:
“To analyze frailty the Hospital Frailty Risk Score (HFRS) was calculated for each patient of the database based on the International classification of disease (ICD)-10 codes at time of admission of the patients using pre-morbid condition of the patients including all data available at the timepoint of stroke admission, including previous admissions. “(92-95)
- ” Univariable logistic regression models were used to identify factors associated with a high impact (p<0.001)” Two grievances here: 1) the p-value does not signify” high impact”, only statistical probability; 2) which parameters were run as univariate models? (and how were multiple comparisons with?
Answer:
1) We changed high impact into statistical probability. (109)
2) The HFRS, TICI-Scale, Elixhauser- as well as Charlston-Comorbitity Index, age, hours ventilation, rate of pneumonia, NHISS at admission and discharge, delta-NIHSS, iv-rtPA, gender, hemicraniectomy, mRS at discharge, ASPECTS, intracranial hemorrhage and time from onset to recanalization were run as univariate models. (111-114)
- Were treatment times (OTT, door-to-groin) available for analysis? It would be surprising if they were not as the data were derived from a prospective stroke registry and known to be important. These should be included.
Answer: We added the onset-to-treatment time and the door-to-groin time into table 2 and 4.
- mRS at admission is not really something one would expect to be reported. The reader will easily confuse this with pre-stroke mRS which it apparently is not here (meadian mRS 5). mRS is not an acute parameter and the authors should stick to NIHSS for this.
Answer: As already remarked by reviewer 2, we excluded the mRS ad admission from our analysis.
- On the other hand, pre-stroke mRS scores would be very important for putting the functional outcome data into context. After all, frailty is strongly associated with functional status and the poor outcome might be partially explained by the already poor functional status before the stroke.
Answer: Unfortunately, regarding the premorbid functional disability, measured by the modified Rankin scale, we only have very incomplete data.
- That could also explain the peculiar dissonance between the results on functional outcome and mortality with frailty being the most important predictor of the first but the least important predictor of the latter in the analyses. Mortality is a far more robust and unequivocal endpoint and also probably closer to the truth in this case considering the subject.
Answer: This is an important point; we agree, that mortality is the far more robust endpoint. We added possible impact of the premorbid functional disability on neurological outcome into the discussion.
“Frail patients have a worse neurological outcome after 90 days than non-frail patients. Only 20% of the medium risk group achieve a favourable outcome after 3 months (mRs 0-2; Fig.1). The effect on neurological outcome might be influenced by premorbid disabilities, since frail patients suffer more often from dementia and premorbid stroke[20] than non-frail patients.” (235-242)
- Since the study period includes both pre- and post-DAWN periods is it possible to analyze patients treated within six hours and patients treated in longer time frames also separately?
Answer: We divided our data into treated within six hours and patients treated in longer time frames. However, we have only 77 patients in the longer time frame and none of them are in the high-risk frailty group. The results of the group treated within six hours do not differ from the whole data set.
- ” However, the results did not reach statistical significance.” Please do not use deterministic wording. It is not a priori clear that the result would be heading that way but just fell short for some reason or other – a point rather nicely emphasised by the behavior of the curve.
Answer: We wrote the sentence a less deterministic.
“No significant difference was detectable in the gradient of the curve.” (175)
- ” The length of stay in the hospital did not differ between 178 patients with a low, moderate or high frailty risk (10 (IQR 11) vs. 10.5 (IQR 8) vs. 9 (IQR 179 5) days; p= 0.656). Moreover, the average length of stay was similar without any significant difference (13.9 (IQR 6) vs. 13.9 (IQR 5) vs. 13.9 (IQR 4); p = 0.763)” This does not make any sense. Please clarify.
Answer: We calculated these items, as in the German DRG System, the compilation of diagnosis are triggering an estimated length of stay in the hospital. Furthermore, the reimbursement is based on the calculated minimum; average and maximum length of stay. We understand, that this rationale might be difficult to follow for the readership outside the DRG system, therefore we removed this sentence: Moreover, the average length of stay was similar without any significant difference (13.9 (IQR 6) vs. 13.9 (IQR 5) vs. 13.9 (IQR 4); p = 0.763)”. (201)
- Indeed, frailty has not been assessed (or at least published) as a predictor of thrombectomy outcome – but advanced age has and, as also the current results show, these tend to go hand in hand. The authors should also discuss previous thrombectomy studies on octo- and nonagenarians. (Btw, there seems to be an Editorial coming up in Age&Ageing soon on the subject)
Answer: We added information about the influence of age, especially nonagenarians into the introduction of our paper.
“It is known, that increasing age is associated with a poor outcome[13]. Octa- and Nonagenarians, treated by mechanical thrombectomy have a higher mortality and morbidity than younger patients. Still, successful recanalization leads to a better neurological outcome and a lower mortality in these patients [17] [18].” (58-61)
- ” But still almost 20% of the medium risk group achieve a favorable outcome after 3 months” So, four out of five patients in this group did not. And the results in the high-risk group were uniformly poor, almost dismal. These points should be emphasised as they are important for clinical decision-making.
Answer: We emphasized on the poor neurological outcome of the medium- and high-frailty risk groups in the discussion.
“Frail patients have a worse neurological outcome after 90 days than non-frail patients. Only 20% of the medium risk group achieve a favourable outcome after 3 months (mRs 0-2; Fig.1) and none of the patients in the high-frailty risk group reached a good outcome. The effect on neurological outcome might be influenced by premorbid disabilities, since frail patients suffer more often from dementia and premorbid stroke[20] than non-frail patients.” (235-242)
- ” Frailty measured by the ICD 10 based hospital frailty risk score is an easy and accessible method, bases on administrative data” I would not call a 109-point list ”easy and accessible” to a clinician. For sure, an EHR appendage could probably calculate it but this leads to the question of the quality of administrative data (as also noted in the discussion) and the method of calculating the score. No, I think these results rather suggest that frailty is also important in thrombectomy patients and should be evaluated clinically.
Answer: As ICD-coding is mandatory for reimbursement, access and calculation of the hospital frailty risk score is easily accessible. We fully agree, that careful coding of complex diagnoses is a time-consuming task. However, once this data is documented with upcoming tools such as the health care card, it will be easier to assess. Our data highlights the usefulness of careful coding beyond reimbursement. We disagree, that frailty should rather be evaluated clinically. We added the possibility of assessing frailty clinically along with potential disadvantages into the discussion. “
“However, clinical frailty scores are often time consuming [26] and have differences in their content validity[27], making frailty difficult to assess.” (263-265)
- Furthermore, its effect seems surprisingly modest here but this might result from the problems with the administrative data already noted? One more good reason to favor clinical assessment and also make a new study along these lines.
Answer: Further studies on the applicability of the HFRS, in emphasis of the quality of administrative data should be performed to decide whether the HFRS can be used in clinical routine or not as well as larger studies of the impact of frailty in a stroke cohort treated with mechanical thrombectomy.
- The style of writing is cumbersome in both the Introduction and the Discussion especially as both are written as singular long paragraphs (the same applies to a lesser extent to Results). These should be revised for the benefit of the reader. Moreover, there are incomplete sentences and repetition. The text should go through a careful round of text editing with fresh eyes.
Answer: We revised the discussion as well as the whole text for incomplete sentences and repetition carefully.
Reviewer 4 Report
In the present study, the authors investigated whether frailty assessed using a dedicated scale mainly based on administrative data influences the outcome in acute ischemic stroke patients aged more than 65 years who were treated by reperfusion techniques. The paper although based on retrospective data is interesting but there are major limitations in the manuscript:
- The HFRS score should be detailed and the main components of the scale presented for readers who are not aware of such type of scale
- A key parameter which is recanalization efficiency is not included in the models although the variation of the NIHSS was, the TICI score after treatment should be however included in the model, we cannot determine whether the results are independent or not of recanalization which might be an important confounding effect; moreover, the authors should also investigate whether frailty at entry influcens the treatment efficiency
- It is not very clear which parameters were kept in the final model for evaluating the predictors of outcomes at 3 months, the list of these parameters should be presented each time
- Since the study was mainly retrospective, it seems that no data was obtained at entry of patients concerning other aspects of neurological status except the Rankin scale, we already know that IQ-Code for evaluating cognitive status is also largely influencing the outcome, the evaluation of previous cognitive and neurological status is only captured using the Rankin scale. This aspect should be either revisited or at least deeply criticized.
- The use of two different comorbidity indices in the multivariate logistic regression models is not explained, do the results are still significant if only one is kept in the model ?
- There are errors in figure 2 about the different categorise of HFRS, the number of individuals in each categories should be also presented
- The difference with previous studies examining frailty as a potential predictor of stroke outcome should be detailed and the novely of the present study clearly presented
Author Response
In the present study, the authors investigated whether frailty assessed using a dedicated scale mainly based on administrative data influences the outcome in acute ischemic stroke patients aged more than 65 years who were treated by reperfusion techniques. The paper although based on retrospective data is interesting but there are major limitations in the manuscript:
- The HFRS score should be detailed and the main components of the scale presented for readers who are not aware of such type of scale
Answer: For a better understanding, we added more detailed information and the main components of the HFRS into the introduction.
“ICD-10 codes with the highest impact are Dementia in Alzheimer’s disease, Hemiplegia, Alzheimer’s disease followed by sequelae of cerebrovascular disease and other signs involving the nervous and musculoskeletal systems, including tendency to fall.” (42-44)
- A key parameter which is recanalization efficiency is not included in the models although the variation of the NIHSS was, the TICI score after treatment should be however included in the model, we cannot determine whether the results are independent or not of recanalization which might be an important confounding effect; moreover, the authors should also investigate whether frailty at entry influences the treatment efficiency.
Answer: We thank the reviewer for this important remark. We added the TICI score into table1 and TICI-scale is part of the regression analysis. Furthermore, we analyzed if frailty affects treatment efficiency. The results are shown in table 4. Moreover, we excluded diagnosis which were due to the stroke, leading to the admission in order to make sure, that we only calculate the frailty prior to admission.
- It is not very clear which parameters were kept in the final model for evaluating the predictors of outcomes at 3 months, the list of these parameters should be presented each time
Answer: The HFRS, TICI-Scale, Elixhauser- as well as Charlston-Comorbitity Index, age, hours ventilation, rate of pneumonia, NHISS at admission and discharge, delta-NIHSS, iv-rtPA, gender, hemicraniectomy, mRS at discharge, ASPECTS, intracranial hemorrhage and time from onset to recanalization were run as univariate models. (111-114)
We added this information into the methods section.
- Since the study was mainly retrospective, it seems that no data was obtained at entry of patients concerning other aspects of neurological status except the Rankin scale, we already know that IQ-Code for evaluating cognitive status is also largely influencing the outcome, the evaluation of previous cognitive and neurological status is only captured using the Rankin scale. This aspect should be either revisited or at least deeply criticized.
Answer: We agree, that this additional information might be interesting. Unfortunately, we cannot provide the IQ-Code from our data. At least a coded diagnosis of dementia was included in the calculation of the HFRS. But we added this issue into the discussion.
“Frail patients have a worse neurological outcome after 90 days than non-frail patients. Only 20% of the medium risk group achieve a favourable outcome after 3 months (mRs 0-2; Fig.1). The effect on neurological outcome might be influenced by premorbid disabilities, since frail patients suffer more often from dementia and premorbid stroke[20] than non-frail patients. Moreover, only the mRS was used to access neurological outcome. Other parameters, influencing the outcome of the patients were not part of this study.” (235-242)
- The use of two different comorbidity indices in the multivariate logistic regression models is not explained, do the results are still significant if only one is kept in the model?
Answer: The results are still significant, when only one comorbidity index is in the model. We changed the multivariate logistic regression model using only the Elixhauser Comorbitiy Index (table 6).
- There are errors in figure 2 about the different categories of HFRS, the number of individuals in each category should be also presented
Answer: We corrected figure 2 and added the number of individuals in each category.
- The difference with previous studies examining frailty as a potential predictor of stroke outcome should be detailed and the novelty of the present study clearly presented
Answer: We added the difference with previous studies to our discussion with an emphasis on the novelty of our work.
Compared to previous studies of frail patients in stroke (18), our study provides a very distinct and highly relevant subgroup of stroke, namely LVO induced stroke, treated with thrombectomy. We applied the HFRS as a well evaluated frailty score, and were able to correct for multiple confunders due to our comprehensive data set." (217-221)
Round 2
Reviewer 1 Report
The criticized points were addressed. My fundamental doubts about the concept of HFRS persist, but the authors' work is solid.
Author Response
Thank you for your valuable review of our paper.
Reviewer 3 Report
Nice work revising. This is almost publishable now but one problem remains.
Namely, this is quite a statement: "We disagree, that frailty should rather be evaluated clinically. We added the possibility of assessing frailty clinically along with potential disadvantages into the discussion."
Naturally, it is alright to disagree but the authors need to make a better case for it or change their conclusion. They do good work in outlining why a register-based method is not realiable for clinical decision-making in individual cases. It is therefore peculiar that they would nevertheless want to rely on it rather than clinical skill. I think this logic does not hold. Furthermore, the point about clinical frailty scales being too cumbersome does no hold water: 10.5770/cgj.23.463
So, either the authors have to explain why incomplete and possibly invalid (nice optimism concerning this in the rebuttal, but the fact remains) administrative data would be better than clinical judgement and the CFS or revise their conclusion. And especially why clinical judgement and the CFS should not be studied in the LVO context.
Author Response
Nice work revising. This is almost publishable now but one problem remains.Namely, this is quite a statement: "We disagree, that frailty should rather be evaluated clinically. We added the possibility of assessing frailty clinically along with potential disadvantages into the discussion."
Naturally, it is alright to disagree but the authors need to make a better case for it or change their conclusion. They do good work in outlining why a register-based method is not realiable for clinical decision-making in individual cases. It is therefore peculiar that they would nevertheless want to rely on it rather than clinical skill. I think this logic does not hold. Furthermore, the point about clinical frailty scales being too cumbersome does no hold water: 10.5770/cgj.23.463
So, either the authors have to explain why incomplete and possibly invalid (nice optimism concerning this in the rebuttal, but the fact remains) administrative data would be better than clinical judgement and the CFS or revise their conclusion. And especially why clinical judgement and the CFS should not be studied in the LVO context.
We thank the reviewer for the valuable comment. We agree, that the recommendation for a specific frailty evaluation needs to be phrased more differentiated and that clinical assessment of frailty is also very valuable. Therefore we revised the corresponding paragraph accordingly:
One advantage of the HFRS is that it can be implemented automatically in hospital information systems. This refers to patients for whom the necessary administrative data are already available at the time of a new stroke admission. Because the ICD-10 code is routinely recorded electronically, determination of the HFRS can be automatically embedded in the hospital's electronic medical record and even has the potential to be programmed into frailty-atuned clinical decision support systems. Having the HFRS automatically available at hospital admission may avoid the challenges of implementing manual scores such as the Clinical Frailty Scale and improving standardization of frailty assessment. This may have additional potential, particularly in stroke patients, where every second counts and acute disability (e.g., hemiplegia, aphasia) may complicate clinical judgement of premorbid frailty status on admission.
Whether the HFRS, a purely clinical frailty assessment, or a combination of both is better suited to predict outcome in frail stroke patients needs to be clarified in further studies. In addition, further studies are needed to determine whether a frailty assessment can guide and optimize clinical care at the individual patient level. (256-266)